# Play to Grade: Testing Coding Games as Classifying Markov Decision Process

**Allen Nie**\*
Computer Science
Stanford University

**Emma Brunskill**
Computer Science
Stanford University

**Chris Piech**
Computer Science
Stanford University

## Abstract

Contemporary coding education often presents students with the task of developing programs that have user interaction and complex dynamic systems, such as mouse based games. While pedagogically compelling, there are no contemporary autonomous methods for providing feedback. Notably, interactive programs are impossible to grade by traditional unit tests. In this paper we formalize the challenge of providing feedback to interactive programs as a task of classifying Markov Decision Processes (MDPs). Each student's program fully specifies an MDP where the agent needs to operate and decide, under reasonable generalization, if the dynamics and reward model of the input MDP should be categorized as correct or broken. We demonstrate that by designing a cooperative objective between an agent and an autoregressive model, we can use the agent to sample differential trajectories from the input MDP that allows a classifier to determine membership: Play to Grade. Our method enables an automatic feedback system for interactive code assignments. We release a dataset of 711,274 anonymized student submissions to a single assignment with hand-coded bug labels to support future research.

## 1 Introduction

The need for high quality education at scale is of critical importance [1]. While delivery of content for millions of students is possible, providing feedback – a cornerstone of education – remains an open challenge. The quality of an online education platform depends on the feedback it can provide to its students. However contemporary coding education has a clear limitation. Students are able to get automatic feedback only up until they start writing interactive programs. When a student authors a program that requires *user interaction*, e.g. where a user interacts with the student's program using a mouse (such as a game), or by clicking on button (such as in a graphical user interface) it becomes exceedingly difficult to grade automatically. Even for well defined challenges, if the user has any creative discretion, or the problem involves any randomness, the task of automatically assessing the work is daunting. This is especially true as coding has become more popular and feedback is required for many students. One popular intro to programming platform, Code.org has over 61 million enrolled students [2]. For their many interactive assignments they have no ability to even identify if a student has a working solution.

Why is providing feedback to interactive programs so hard? The standard way to provide feedback is through human labor. Teachers need to interact with each student's program for 20 seconds to 10 minutes in order to grade. This may seem small, but for the quantity of unique programs on a platform such as Code.org that amounts to approximately 9.5 years of human effort to provide feedback for a single assignment. Code.org has tried crowdsourcing feedback from hundreds of thousands of teachers which has fallen short for similar reasons: labelling is hard and undesirable work [3]. Interactive assignments can not be auto graded as running the program requires *dynamic*

---

\*anie@stanford.edu

35th Conference on Neural Information Processing Systems (NeurIPS 2021).

input. Moreover, heavily scaffolded assignments – ones that force students to program their interactive programs in a way that enables standard testing – are considered to be a sub-par learning experience for students and are not common in contemporary classrooms.

There is a long history of work towards providing feedback to students working on open ended assignments which has shown that automatically providing feedback based on code text is a hard machine learning problem [4, 5, 6, 7, 8, 9]. This is true for many reasons. Even for introductory level computer science education, homework datasets have statistical distributions with heavy tails similar to natural language [10]. Moreover, these distributions tend to be highly discontinuous – two solutions which are only slightly different in text can be very different in its behavior. As such, approaches to grading which rely only on reading the text of a students code end up being as complex as understanding a passage of natural language [11]. Finally, because feedback is necessary for the first student working on an assignment, not just the millionth, and new assignments are often introduced, the challenge is inherently few shot.

In this work we show that an algorithm that learns to interact with a student's assignment, to *play* with the student work, can enable new capacity for understanding student work. A collaborative system which can simultaneously learn to understand what type of behavior is undesirable, as well as play student's work to actively trigger these undesirable behaviors, is the crucial first step to developing automatic and intelligent feedback for massive online coding education.

**Our contributions:**

- We introduce the reinforcement learning challenge of *Play to Grade*.
- We propose a collaborative algorithm where an agent simultaneously learns to play a game and recognize what states are bug inducing states.
- Our solution obtains 93.4%-94.0% accuracy on a real-world coding dataset provided by Code.org. We gained a 19-25 percentage point improvement over grading programs via code text.
- We release a Bounce dataset of student submissions with ground truth bug labels and an OpenAI-Gym compatible environment to support further research: `https://github.com/windweller/play-to-grade`.

### 1.1 Related Work

**Education feedback**  The quality of an online education platform depends on the feedback it can provide to its students. Low quality or missing feedback can greatly reduce motivation for students to continue engaging with the exercise [12]. Currently, platforms like Code.org that offers block-based programming use syntactic analysis to construct hints and feedbacks [13]. Using recurrent neural networks to understanding programs through code text has also been well-explored, focusing on providing code-level feedbacks or correcting syntactical errors [14, 15]. Other works generate hints based on writing complicated automatic syntax tree [16, 17], which could place heavy burden on teachers and education platforms. The current state-of-the-art introduces a method for providing coding feedback that works for assignments up to approximately 10 lines of code [10]. The method does not easily generalize to longer and more complicated programs. We take inspiration that the underlying code can vary but the correct behaviors should all be the same. Our method sidesteps the complexity of static code analysis and instead focus on analyzing the MDP specified by the game environment. Our work is complementary to these methods because we provide a new tool of analysis for understanding a program's behavior and providing feedback beyond static text analysis.

**Reinforcement learning for software testing**  Designing automated test for software has been well-explored by using template-based test case generation [18] and input fuzzing [19]. Recently, there have been a lot of interests in using RL agents to find bugs in games. The most prevalent solution is an agent that heavily focuses on exploring new states [20, 21, 22, 23]. Such agent optimizes an objective such as discretely counting the number of states the agent can reach [24] or use curiosity objective to encourage visiting new states [25, 26]. They all rely on the strong assumption that once a bug state is reached, we will be able to know that the state is a bug. We show that exploration (or reaching the bug state) is only half of the challenge. However, it is possible that these pure-exploration agents or input fuzzing can provide additional diverse trajectories to further optimize the bug classifier. The idea of pairing a RL agent with a predictive model is also explored in safe RL, such as predicting whether a state is unsafe [27, 28] and training agents to seek out unsafe states [29, 30].

## 2   The Play to Grade Challenge

In this section we are going to introduce a novel challenge for the machine learning community: how can an algorithm learn how to "play" with a student submission in order to understand the ways in which it might be buggy? The challenge applies to the task of giving feedback for any interactive student work. In this paper we will focus on games typical of introduction to programming courses.

We formulate the challenge with constraints that are often found in the real world. Given an interactive coding assignment, a teacher often has a few solution implementations of the assignment. We also assume that the teacher can prepare a few incorrect implementations that represent their "best guesses" of what a wrong program should look like. This sets up a few shot challenge where an algorithm must learn to generalize from the few teacher examples, to the tens-of-thousands (or more) unlabeled combinations of mistakes and invariances found in natural student code.

**Definition 2.1.** *A deterministic **Markov Decision Process (MDP)** is a 4-tuple M = <$\mathcal{S}, \mathcal{A}, T, R$>, where $\mathcal{S}$ is a set of states; $\mathcal{A}$ is a set of actions; $T : \mathcal{S} \times \mathcal{A} \to S$ is the transition dynamics; and $R : \mathcal{S} \times \mathcal{A} \to \mathbb{R}$ is the reward function.*

We can consider each student implementation of the interactive assignment as a separate MDP. Shared structures between these MDPs can be leveraged to train a grading algorithm. The first assumption that we take is that all MDPs share the same action space $\mathcal{A}$. This is not a difficult requirement to satisfy because specifying how a game should be played is often part of the instruction for the homework. The second assumption is that all MDPs in this set share the same state space $\mathcal{S}$. This can be satisfied in two ways: 1) If the algorithm takes game objects' positions and dynamics as input, we can design the state space to include as much information as possible (i.e., grab all objects on the canvas); 2) If the algorithm takes screenshots/images of game play, then the state space is trivially the same, up to a scaling factor.

To formalize this setting, we consider a *training* dataset, $\mathcal{D}$, of $n$ programs, each fully specifies an environment and its dynamics: $\mathcal{D} = \{(M_n = (\mathcal{S}, \mathcal{A}, T_n, R_n), y_n) : n = 1, 2, 3, ...\}$, where $y_n \in \{0, 1\}$ is a binary label for the MDP $M_n$, and a set of programs similarly specified to be tested in $\mathcal{D}_{\text{test}}$. We note that $n$ is often a small number, usually less than 20, significantly fewer than other supervised learning tasks. $y = 0$ means the game has no bug, and $y = 1$ means the game has at least one bug. The training objective is to correctly assign a label $\hat{y}_n$ for each $M_n$. The play-to-grade challenge is to show that a system trained on this few-shot training data can be generalized to accurately identify bugs for all students in the class. Though our work focuses on the binary bug labeling, our proposed algorithms are not inherently limited by this. We leave multi-class bug labeling (i.e., which bug is discovered in the MDP) for future work and note they can provide even larger impacts on generating fine-grained feedbacks to student assignments.

### 2.1   Equivalence Relations in MDP

The study of equivalence relations in MDPs and labelled transition systems has a long history. Previous works focus on determining equivalence relations of states in order to condense state representations for easier policy learning [31, 32, 33, 34, 35]. The general notion is that if two states are equivalent, they should receive similar reward and lead to a similar next state. Therefore, a distance measure $d$ can be defined to judge whether two states are equivalent.

To solve the play to grade challenge, instead of considering whether two states are equivalent, we can try to determine whether the two MDPs are the same. If the reference MDP has reward function $R$ and transition function $T$, we only need to know if $\tilde{T}$ and $\tilde{R}$ from the new MDP are the same as the reference MDP. Unfortunately, since we cannot directly compare the parametrized functional forms, we extend the distance metric developed by [33] from state equivalence relations to MDP equivalence relations by sum over all state in Eq 1. We can use any norm to measure distance of next state for continuous state space, and use an indicator function if the state is discrete. We can use $\alpha$ and $\beta$ to decide how much we weigh the difference between MDPs.

$$D(M, \tilde{M}) = \sum_{s \in \tilde{\mathcal{S}}} \max_a \alpha |R(s, a) - \tilde{R}(s, a)| + \beta \|T(s, a) - \tilde{T}(s, a)\| \tag{1}$$

However, we will not be able to compute $D(M, \tilde{M})$ exactly for non-tabular MDPs because we cannot efficiently enumerate over all possible states. Our goal is to find some kind of difference between the two MDPs so that we can confidently reject the notion that the given MDP $\tilde{M}$ is the same as our

reference MDP $M$. Therefore, instead of calculating the difference between the two MDPs over all states, we only need a subset of states where the difference is the most significant. As long as we can discover a subset of states, or even one state where a significant difference occurred, we can decide that two MDPs are not the same. We call these states the **differential states**. The difficulty of The challenge is determined by the cardinality of the differential states.

## 2.2 Play to Grade Objective

The objective is to reach differential states so that we can confidently determine if the student solution is sufficiently different from a reference MDP: $D(M, \tilde{M}) > \nu$, where $\nu$ is chosen by the teacher, and can be thought of as some fault tolerance level. Because $D(M, \tilde{M})$ is a summation over norms, identifying some or any differential states would trivially satisfy this objective. We can rewrite this existential search objective as an optimization instead. We can define a new MDP where the reward function is the difference on $(s, a)$, and the rest are the same as $\tilde{M}$. We can obtain a policy that learns to maximize this reward, which is equivalent to searching for differential states by sampling **differential trajectories** between two MDPs:

$$d(s, a) = \alpha |R(s, a) - \tilde{R}(s, a)| + \beta \|T(s, a) - \tilde{T}(s, a)\| \tag{2}$$

$$\pi = \arg\max_\pi \mathbb{E}_{(s,a) \sim \pi} \Big[ \mathbb{1}(d(s, a) > \delta) \Big] \tag{3}$$

Now that we have a clear optimization objective that we can solve it using any reinforcement learning algorithm, we must consider another difficulty. Even though the MDP framework provides a simple abstraction to the problem we face, we do not have real access to the underlying reward and transition function. Given a state $s$ from $\tilde{M}$ and action $a$ from $\pi(s)$, we are unable to directly query the reference MDP $M$ for $T(s, a)$ and $R(s, a)$. This is because we are in an episodic setting, where the direct access to transition function and reward function is not possible. Also, we generally assume that each episode will have a random initial state $s_0 \sim p(s_0)$ (i.e., in breakout, the ball gets launched at a different downward direction each time you play). This randomness is defined internally and does not allow for outside control (such as seeding). This means we can't simply use one policy $\pi$ to sample two trajectories from both MDPs and directly compare the reward and the next state.

## 3 Recognizing Bugs

The main challenge of solving this task is: if we know what differential states look like (i.e., have a good state distance function $d(s, a)$), we can directly train an agent to reach these states; conversely, if we have a good agent that can reliably reach these differential states all the time, we can find a good $d(s, a)$. The fact that we have neither, is the *cold-start* problem of this challenge.

We will address the *cold-start* problem in the next section, but first, if we assume that we do have a good policy that can generate differential trajectories, how should we parametrize and learn a distance function $\hat{d}(s, a)$ that recognizes $(s, a)$ does not belong to the reference MDP? We introduce a few baselines and propose two methods that have a good inductive bias to recognize differential states.

### 3.1 Baseline: Noisy Supervised Learning

A good distance function can be learned with noisy supervised learning. Since we have a small set of labelled MDPs (i.e., $(M, y = 0), (M, y = 1)$) in our training dataset $\mathcal{D}$, we can label all state-action tuples sampled from the correct MDP as no-bug, and all state-action tuples from the incorrect MDP as bug, and train a supervised classifier. We define the state classification function that we use to label state-action tuples as $\phi : \mathcal{S} \times \mathcal{A} \times \mathcal{S} \to \{0, 1\}$. $\phi(\tilde{s}, \tilde{a}, \tilde{s}') = 0$ if $(\tilde{s}, \tilde{a}, \tilde{s}')$ are sampled by evaluating $\pi$ on $M$, otherwise they are sampled from $\tilde{M}$.

### 3.2 Baseline: Unsupervised Learning

Another baseline idea is to learn the distribution of $(s, a)$ sampled from $M$. We can train a generative model on $(s, a, s')$ triples collected from $(M, y = 0)$. The model can optimize to approximate a joint distribution $p_\theta(s, a, s')$. We compare a Gaussian mixture model and a variational autoencoder [36] as our generative model of choice. We turn our generative model into a classification function, using the same $\phi$ from Section 3.1: $\phi(\tilde{s}, \tilde{a}, \tilde{s}') \triangleq p_\theta(\tilde{s}, \tilde{a}, \tilde{s}') \geq \sigma$. For both baseline algorithms, we can

assign a label to the new MDP $\tilde{M}$ by computing $\hat{D}(M, \tilde{M}) = \mathbb{E}_{(s,a)\sim\pi,\tilde{M}}[\phi(s, a, \tilde{T}(s, a))]$ and set a decision threshold $\hat{D}(M, \tilde{M}) \geq \nu$.

## 3.3 HoareLSTM

Since the policy samples sequential observations from an MDP, it is natural to calculate distance as a form of unexpectedness between predicted state and observed state. In fact, the programming language community has been looking into a triple of pre/post-condition along with an action input for a long time. These are called *Hoare triples* [37]. They have also been extended to measure state equivalence in MDP [38]. Previous work in computational education has used them to learn program embeddings for generating feedbacks [14].

We can approximate the distance function $d(s, a)$ in Eq 2 by learning a transition and reward model. Unlike model-based reinforcement learning [39], where an accurate model of transition model $T$ and reward function $R$ must be learned for policy optimization, we only need to optimize the models to the extent that we can confidently make decisions about $\tilde{M}$'s label.

As explained before, we only have access to either $(T, R)$ or $(\tilde{T}, \tilde{R})$, but not both. We can learn a predictive model $R_\theta(s, a)$ and $T_\theta(s, a)$ for the reference MDP's $(T, R)$. We note that the model does not need to be autoregressive, but we choose to parametrize them as an autoregressive long-short-term memory (LSTM) network [40]. We choose $f$ and $g$ to be fully connected neural networks with separate parameters. Since we are predicting Hoare triples, we call this the HoareLSTM model.

$$h_t = \text{LSTM}(s_t, a_t, h_{t-1}) \tag{4}$$

$$\hat{s}_{t+1} = f(h_t); \hat{r}_t = g(h_t) \tag{5}$$

$$\mathcal{L}(\theta) = \mathbb{E}_{(s,a)\sim\pi,M}\left[|\hat{r}_t - R(s, a)| + \|\hat{s}_{t+1} - T(s, a)\|\right] \tag{6}$$

If we learn $T_\theta$ and $R_\theta$ to approximate $T$ and $R$ from the reference MDP $M$, we can compute a state-action tuple distance function from $\tilde{M}$ to $M$, given any new MDP $\tilde{M}$. We denote this approximate distance function as $\hat{d}_M$ that takes $\tilde{T}$ and $\tilde{R}$ as input:

$$\hat{d}_M(s, a; \tilde{T}, \tilde{R}) = \alpha|\hat{r}_t - \tilde{R}(s, a)| + \beta\|\hat{s}_{t+1} - \tilde{T}(s, a)\| \tag{7}$$

$$\hat{D}(M, \tilde{M}) = \mathbb{E}_{(s,a)\sim\pi,\tilde{M}}\left[\hat{d}_M(s, a)\right] \tag{8}$$

We use $\hat{D}(M, \tilde{M})$ to compute the difference between the predicted trajectory by approximating $T$ and $R$ from MDP $M$ and observed trajectory by actually sampling from $\tilde{T}$ and $\tilde{R}$. At last, we can determine if $\tilde{M}$ is the same as $M$ by checking if $\hat{D}(M, \tilde{M}) \geq \nu$.

## 3.4 Contrastive HoareLSTM

The distance we calculate is highly reliant on how well our model can approximate the system dynamics of state evolution. However, our model has inherent approximation error. We present the idea of a contrastive distance to potentially mitigate this error.

Instead of learning to approximate one reference MDP's $(T, R)$, we can learn to approximate a correct MDP $M^+$ with $(T^+, R^+)$, and a bug MDP $M^-$ with $(T^-, R^-)$. Given a new test MDP's $(\tilde{T}, \tilde{R})$, we can perform a contrastive comparison to determine the distance function $\hat{d}_{M^+, M^-}$:

$$\hat{d}_{M^+, M^-}(s, a; \tilde{T}, \tilde{R}) = \alpha|\tilde{R}(s, a) - \hat{R}_\theta^+(s, a)| - \alpha|\tilde{R}(s, a) - \hat{R}_\theta^-(s, a)| + \tag{9}$$

$$\beta\|\tilde{T}(s, a) - \hat{T}_\psi^+(s, a)\| - \beta\|\tilde{T}(s, a) - \hat{T}_\psi^-(s, a)\| \tag{10}$$

We still use policy $\pi$ to sample $(s, a)$ from the new input MDP $\tilde{M}$, but instead of directly comparing it with one MDP $M$, we are now comparing it with two MDPs $M^+$ and $M^-$, and see which transition and reward model better predicts the behavior of the new input MDP.

The main advantage of using a difference comparison is that we are likely to be more fault tolerant. If we look at how we decide if a MDP has a bug: $\hat{D}(M, \tilde{M}) \geq \nu$, the choice of $\nu$ needs to reflect our approximation error when we try to learn $T$ and $R$ so that we are not classifying an correct MDP as a bug MDP if our model is not able to recreate the transition dynamics accurately. If we approximate both $(T, R)$ and $(\tilde{T}, \tilde{R})$ and use the same family of models with similar parameter size,

the approximation errors could cancel out in the contrastive setting. The disadvantage is that now we need to train two models, and the distance can be influenced by how well each model is optimized.

**Algorithm 1:** Collaborative Reinforcement Learning

TRAINDIFFERENTIALPOLICY
$(M, \tilde{M}, \pi_0, \hat{d}_\theta)$
$\delta = 0.1$
**for** $n \leftarrow N$ **do**
 $\mathcal{D}_\tau = $ CollectTrajectory$(M, \tilde{M}, \pi_0)$
 $\hat{d}_\theta = $ TrainDistFunc$(\mathcal{L}(\theta), \mathcal{D}_\tau)$
 $\delta = $ AdjustDelta$(M, \tilde{M}, , \pi_0, \delta, \hat{d}_\theta)$
 **Let** $r^\star = \mathbb{1}(\hat{d}_\theta(s, a) > \delta)$
 $\pi_n = $ TrainDQN$(\pi_0, \tilde{M}, r^\star)$
**end**

**return** $\pi_N, \hat{d}_\theta, \delta$

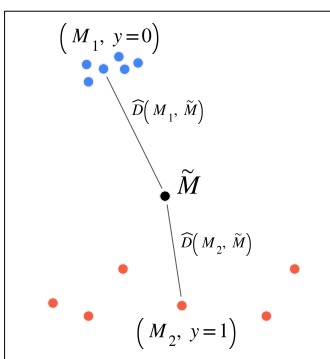

Figure 1: Grading interactive coding assignments as pairwise distance comparisons between few-shot reference MDPs.

## 4  Collaborative Training

With a parametrized approximate distance function $d(s, a)$ and an agent $\pi$ that learns to maximize this distance function as reward, the solution seems complete. However, neither the distance function nor the agent can work well in the beginning. Agent acts randomly on the states, and the distance function can't capture any difference between states. This is the *cold-start* problem: we need to simultaneously learn to play the game and find bugs. We present some standard agents and leave more detailed investigations to future work.

### 4.1  Initial Agent: Random or Play-to-Win

One solution is to start with an agent that acts randomly. This strategy is particularly useful if the state space can be easily explored. By sampling random trajectories, we might be able to quickly train a good distance function. The drawback is the inefficiency and the fact that a random agent will struggle with games that have hard-to-reach states.

Another solution is to train a policy to maximize the original reward (i.e., play to win). If the reference MDP has a clearly defined reward function that gives reward, we can train a policy $\pi$ that simply learns to maximize the episodic total reward. This is a very useful pre-training to hot-start the policy, especially important in games with long horizon in which it is easy to fail.

The problem with this approach is that the policy is biased toward finishing the original objective in the coding game. If the bug states can only be reached by losing the game, it will not lead to any bug discovery. Another problem is that some coding games do not have clearly defined reward. In that situation, we can use curiosity based objective [21]. But as we will show in the experiment section, play-to-win policy is often good enough at sampling sufficient differential trajectories to train a good distance function.

### 4.2  Improved Agent Using Iterative Training

With the initial agent, we can start to collect trajectories that we can use to train the underlying parameters of our distance function. If our parametrized distance function is a discriminative model (i.e., noisy supervised learning), the optimality of the agent will have an impact on the quality of state level pseudo labeling. We can see in Section 5.1.1 and Section 5.1.2, noisy supervised learning can do quite well as long as the agent can reliably reach the bug states. If our distance function is a generative model that estimates the joint distribution of the observed data, how the agent collects trajectories does not affect the learning of the model. However, we still need to set a threshold $\delta$ to determine if a $(s, a)$ is a bug-inducing state or not, as indicated in Equation 3.

Empirically, for our environments, we find that we do not need to more than 1 iteration of collaborative training. Therefore, we are able to adjust $\delta$ at the end of each iteration manually by looking at the distance-to-self $\hat{D}(M, M)$ – if we take $(s, a)$ from the reference MDP and use $\hat{d}$ to calculate the difference to itself, we can choose a $\delta$ so that all $(s, a)$ can be labeled as correct. We leave the exploration of how to automatically adjust $\delta$ to future work and more complicated environments.

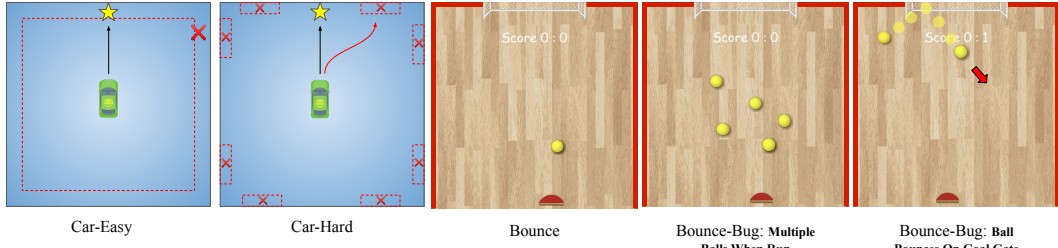

| Car-Easy | Car-Hard | Bounce | Bounce-Bug: **Multiple Balls When Run** | Bounce-Bug: **Ball Bounces On Goal Gate** |

Figure 2: We show the illustration of the car environment and two settings on the left, and bounce environment and its two incorrect implementations on the right. For the car environment, bug states are indicated by the dotted red lines and the error signs inside. Note that the correct car environment only has the car and the gold star, without any dotted line or error sign.

## 5 Experimental Results

### 5.1 Car Environment

We use a toy example to explore some basic aspects about the play-to-grade challenge. In this toy example, we continue our intuition that the goal is to reach and identify the bug-triggering state. We build a 2D environment where the agent needs to drive a car to reach a gold state, and potential bug states are defined as specific coordinates on this 2D field. All states are easily accessible by the agent. This is the most generalized and simplest representation of the play-to-grade challenge. We can use this toy environment to help us understand some fundamental challenges in play-to-grade and understand the properties of collaborative training.

In all car environments, the agent can finish the game by reaching the gold star. When a bug-state is reached, the car's physical movement is altered, resulting in back-and-forth movement around the same location – reflecting the idea that the car is "stuck".

#### 5.1.1 Bug Classification

We define a Car environment setting: Car-Easy, where almost all agents can easily reach the bug state regardless of whether they adopt the collaborative training objective or not (unless this agent is adversarial). The objective of this setting is to test how well bug classifiers perform with an optimal agent. Although most agents can be optimal, they are not equivalent in the eye of the bug classifiers, because the trajectories these agents collect will differ quite significantly. We test two agents, one that only drives a straight line to the gold star state (the single-direction agent); the other drives randomly (the random agent). The trajectory generated by a random policy is much harder to learn to predict and memorize compared to a single direction policy.

In Table 1, all models including baseline models are able to predict bug states with high precision in a single direction agent setting. However, only our proposed methods: HoareLSTM and Contrastive HoareLSTM are able to learn well by predicting next states in an autoregressive manner. We report the model specification and optimization details in appendix.

#### 5.1.2 Collaborative Training Improvement

Most of the time, we will not have an optimal agent that can always reach a bug state in every trajectory. Therefore, it is important for us to test bug states that are sparse and slightly harder to reach. We devise the Car-Hard setting where bugs only appear in small rectangular regions of the 2D plane. This creates a situation where the agent only encounters bug states if the agent plays into the bug state. We use this to show that letting the classifier and the agent train iteratively can improve the performance of both classifier and agent. In Table 2, we show that by applying collaborative training (CT), we allow the agent to become much better at reaching bug states, and in return, allow us to train a much stronger bug classifier even after one iteration. We argue that CT is necessary when bug states are far away from the goal and that is the reason why recall is increased on most models after CT.

### 5.2 Bounce

**Data source** *Code.org* is an online computer science education platform that teaches K-12 students beginner programming using a drag-and-drop interface. Our dataset is compiled of 453,211 students

| | Single Direction Agent | | | Random Agent | | |
|---|---|---|---|---|---|---|
| | Accuracy | Precision | Recall | Accuracy | Precision | Recall |
| **Baselines** | | | | | | |
| Noisy Supervised Learning | $91.0 \pm 4.7$ | $90.3 \pm 5.1$ | $98.6 \pm 1.8$ | $60.3 \pm 11.8$ | $48.0 \pm 14.8$ | $84.6 \pm 17.8$ |
| Gaussian Mixture Model | $25.3 \pm 12.5$ | $60.6 \pm 26.0$ | $23.8 \pm 15.7$ | $17.0 \pm 10.6$ | $18.0 \pm 11.0$ | $7.3 \pm 6.8$ |
| Variational Autoencoder | $83.8 \pm 4.4$ | $84.9 \pm 5.0$ | $95.7 \pm 1.1$ | $69.5 \pm 6.4$ | $51.6 \pm 10.9$ | $86.0 \pm 8.5$ |
| **Our methods** | | | | | | |
| HoareLSTM | $99.3 \pm 1.9$ | $99.0 \pm 2.7$ | $100.0 \pm 0.0$ | $98.5 \pm 1.1$ | $99.6 \pm 1.0$ | $94.1 \pm 6.0$ |
| Contrastive HoareLSTM | $100.0 \pm 0.0$ | $100.0 \pm 0.0$ | $100.0 \pm 0.0$ | $94.5 \pm 1.8$ | $94.4 \pm 2.8$ | $86.0 \pm 5.2$ |

Table 1: Bug-state level result of the Car-Easy environment. We report the result of both agents over 5 runs. For non-sequential models, we stack the previous 4 states with the current state. We measure the accuracy of correctly classifying bug or non-bug states. Recall and precision are measured only for bug states. We report 95% confidence interval.

| | Random Agent | | | CT - Iteration 1 | | | CT - Iteration 2 | | |
|---|---|---|---|---|---|---|---|---|---|
| | Acc | Prec | Rec | Acc | Prec | Rec | Acc | Prec | Rec |
| **Baselines** | | | | | | | | | |
| Noisy Supervised Learning | 46.8 | 12.6 | 37.2 | 67.4 | 53.6 | 94.0 | 67.7 | 47.7 | 82.6 |
| Gaussian Mixture Model | 8.4 | 3.4 | 2.1 | 11.8 | 8.3 | 2.9 | 12.3 | 0.0 | 0.0 |
| Variational Autoencoder | 63.8 | 32.0 | 96.4 | 66.5 | 53.0 | 95.6 | 77.8 | 74.8 | 96.7 |
| **Our methods** | | | | | | | | | |
| HoareLSTM | 99.2 | 76.5 | 80.0 | 100.0 | 100.0 | 100.0 | 99.3 | 99.0 | 100.0 |
| Contrastive HoareLSTM | 98.4 | 93.9 | 87.1 | 97.2 | 99.7 | 94.9 | 97.9 | 99.9 | 96.6 |

Table 2: Bug-state level result of the Car-Hard environment. We show how collaborative iterative training can help classifiers get higher performance. We also average over 5 runs. We measure the accuracy of correctly classifying bug or non-bug states. Recall and precision are measured only for bug states.

who wrote a solution to the Bounce assignment. In total, there are 711,274 submissions, where 111,773 unique programs were submitted. Since the data are student programs, there is no personally identifiable information in our dataset. All programs have been manually labeled with the set of bugs they contain. We provide a data risk statement in Section A.5.

**Assignment detail** On Code.org students use a drag-and-drop style code editor to program the game of Bounce, a single player version of Pong where a player tries to bounce a ball into a goal using a paddle. Students program if/then relationships such as "when run", "launch ball". The "launch ball" command shoots a ball in a random direction and the randomization is not editable by the agent. There is no limit on how many commands can be used, allowing freedom to, for example, add five "Launch new ball" under one condition. Two example bugs are visualized in Figure 2. Bounce was intentionally chosen as it was difficult for an agent to autonomously grade based on playing, but easy for human annotators to provide gold-standard bug-labels based on looking at the source code.

**Evaluation** In an unbounded solution space, the distribution of student submissions incur a heavy tail, as observed by [10]. We show that the distribution of submissions in dataset conforms to a Zipf distribution Figure A.3b. This suggests that we can partition this dataset into two sections. **Body**: we count any unique program submitted by more than 10 students as the "body" of the distribution. It accounts for 80.0% (565,714) of the total submissions. This set contains 500 unique correct programs and 2,690 unique incorrect programs. **Tail**: This set represents any programs that are submitted less than 10 times. This set contains 101,986 unique incorrect and 6,597 unique correct solutions. For both Body and Tail distribution, we sample 250 correct and 250 incorrect programs uniformly from each set for evaluation.

|  | | Correct Program | | Broken Program | |
|  | Accuracy | Precision | Recall | Precision | Recall |
|---|---|---|---|---|---|
| **Body Student Programs (566K submissions, 3,190 unique)** | | | | | |
| Majority Class | 50.0 | — | — | — | — |
| Code-as-text | 74.2 | 84.6 | 59.2 | 68.6 | 89.2 |
| Contrastive HoareLSTM + PTW | 93.4 | 88.6 | 99.6 | 99.5 | 87.2 |
| **Tail Student Programs (146K submissions, 108,583 unique)** | | | | | |
| Majority Class | 50.0 | — | — | — | — |
| Code-as-text | 68.4 | 85.9 | 44.0 | 62.4 | 92.8 |
| Contrastive HoareLSTM + PTW | 94.0 | 91.0 | 97.6 | 97.4 | 90.4 |

Table 3: We report the pre-trained "play-to-win" (PTW) agent's performance when predicting if a program has a bug with different methods of bug identification. For both body and tail distribution, we sample 250 correct and 250 broken programs for a balanced evaluation.

**Analysis**    We evaluate our algorithm's ability to find bugs in student Bounce programs. Unlike the Car environment, Bounce has most bug states close to the goal state; for example, if the wall or paddle does not bounce the ball, a game cannot be complete. Therefore, we focus on evaluating our bug prediction. We train our prediction on 10 incorrect programs and 1 correct program. We report the result of Contrastive HoareLSTM trained with trajectories sampled by a pre-trained play-to-win agent Bounce. The final grading label is determined by a majority vote with 10 contrastive distance functions. Table 3 shows that we are able to achieve high accuracy (93.4-94.0%) relative to the code-as-text baseline on the binary classification of student assignment. This shows the potential of play-to-grade algorithms in assisting in real-world classrooms.

## 6   Discussion

**Providing helpful feedback**    Our current formulation of "identifying a bug state" holds tremendous educational utility. For beginner programmers, it is important for them to see where the program went wrong and try to develop a solution by themselves. By identifying a bug state, we can record a 2-3 seconds short video around the bug state and display to students where their program made a mistake. We plan to collaborate with Code.org to deploy our trained algorithm in a real production environment, but this live experiment is beyond the scope of our technical paper.

**What if game internal states can't be retrieved?**    There are many situations where the internal object representations (such as velocity, position) can't be accessed easily from outside; in this case, often all the information we have would be the image snapshots of the game state. It is true that the iterative learning paradigm we proposed would be more compute intensive, but the model components we explored in this paper are capable of learning and operating on pixels [41].

**Does this work for non-game programming assignments?**    Many interactive student assignments might not be a "game" and the reward might not be clearly defined. However, since our algorithm does not rely on the internal reward of the game – we only rely on the classifier to discover differential trajectories between correct and incorrect programs, our solution should generalize to other types of programming assignments that require interactive user inputs such as web apps.

**Creativity as a distance**    Often times, students create solutions that are creative. Understanding and rewarding creativity is the major challenge in AI for education. Though we didn't use the Bounce dataset to focus on the problem of understanding creativity, our work opens up an interesting angle to address this very hard challenge. In this work we identified several unique approaches that we believe will be useful in recognizing creativity. The idea of play-to-grade could help to identify the difference between a truly broken solution and one which is playable, but different from the reference solution. Or perhaps we could think of creativity as being related to the delta (distance to the correct solution). We imagine that this line of research will be very exciting future work for both the fields of education and reinforcement learning.

**Future work** We see several promising directions to extend this work. First, given that Bounce dataset includes multi-class bug labels, turning the binary classification into a multi-class multi-label classification would bring some new exciting challenges to our framework. Second, as shown in Table 2, iterative training is not always monotonically improving the performance of the algorithm. Exploring the properties of iterative training is left for future work.

## 7    Conclusion

Providing a generalizable algorithm that can play interactive student programs in order to give feedback is an important problem for education and an exciting intellectual challenge for the reinforcement learning community. In this work, we introduce the challenge and a dataset, set up the MDP distance framework, provide algorithms that achieve high accuracy, and demonstrate this is a promising direction of applying machine learning to assist education.

## Acknowledgement

This work was supported in part by a Stanford Hoffman-Yee Human Centered AI grant. We would like to thank code.org, and Baker Franke, for generously providing the research community with data. We additionally thank Lisa Yan, Tong Mu, Mike Wu, Yunsung Kim, Ali Malik, Moussa Doumbouya, Henry Zhu for feedback on this project.

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
