# A   Appendix

## A.1   HoareLSTM and Contrastive HoareLSTM Training Data

Two algorithms presented in Section 3.3 and Section 3.4 require different trianing data. HoareLSTM only requires one reference program that is correct. However, Contrastive HoareLSTM requires the teacher to provide a few (in the case of Car, one; in the case of Bounce, ten) incorrect programs. We believe this does not place a heavy burden on the teachers.

## A.2   Car Environment

Let the center of the environment be $(0, 0)$, the opposite boundaries are -10 and 10. The car's initial x-y coordinates are uniformly sampled from $[-2, 2]$. There are four discrete actions for the agent to take, each action applies an acceleration or deceleration of 0.2 to the velocity along the corresponding direction, with max speed of any direction capped at 1. This allows the car to have more complex trajectories than zig-zag lines.

When a bug-state is reached, the car's physical movement is altered. The velocity is first clipped to be in $[-0.5, 0.5]$. At each step, instead of responding to an action from the agent, the car simply flips the sign of the previously clipped velocity, resulting in back-and-forth movement around the same location – reflecting the idea that the car is "stuck".

## A.3   Car-Easy Experiment Setup

For HoareLSTM and contrastive HoareLSTM, we use a fully connected network (FCN) to project input into 128-dim latent representations, and then use a LSTM with hidden state dimension of 128, and at last, use a FCN to project it down to output. The nonlinearity is GeLU (Gaussian Error Linear Units function).

## A.4   Car-Hard Agent Trajectory

We can directly examine how collaborative training taught the agent by looking at its trajectory. At first, the agent drives the car randomly. But after 1 round of collaborative training, the agent becomes sharply focused and only visits the possible buggy areas.

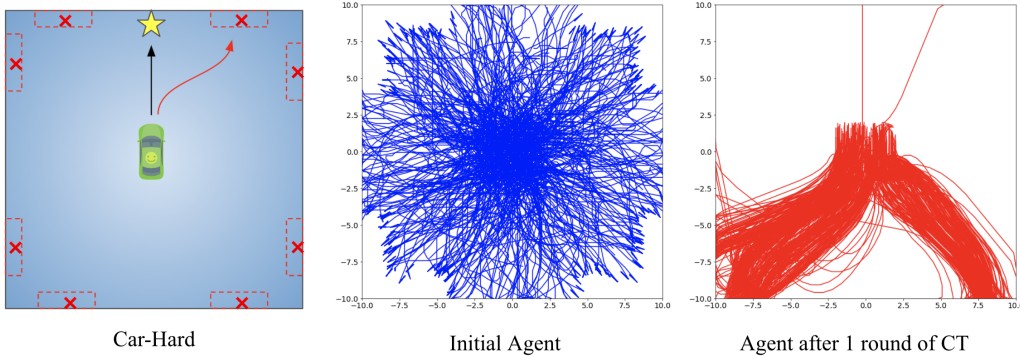

Car-Hard                     Initial Agent                     Agent after 1 round of CT

Figure A.1: We show in the Car-Hard environment, a random agent (in blue) can use bug classifier's decision as reward to quickly learn to drive straight to the bug states (in red).

## A.5   Bounce Data Risk Statement

As shown in Figure A.3a, because Bounce is only a drag-and-drop interface, there is no place to add custom comments or include any other custom text in the homework submission. The dataset we release does not provide timestamp, identifier information, and any metadata linked to the submission. The dataset only contains the programs themselves, which are represented in JSON. We deem the risk of exposing personal identifiable information through our dataset to be negligible.

## A.6 Bounce Task Details

**Gold annotations**   We generate the ground-truth gold annotations by defining legal or illegal commands under each condition. For example, having more than one "launch new ball" under "when run" is incorrect. Placing "score opponent point" under "when run" is also incorrect. Abiding by this logic, we put down a list of legal and illegal commands for each condition. We note that, we intentionally chose the bounce program as it was amenable to generating gold annotations due to the API that code.org exposed to students. While our methods apply broadly, this gold annotation system will not scale to other assignments. The full annotation schema is provided as code in the code base.

**Interface**   We show an interface of how students write the program in Figure A.3a. We also show a log-log distribution plot to show that the distribution of unique student programs conforms to a Zipfian distribution. Even though students can create visually stimulating programs by setting various themes (illustrated in Figure A.2), we side-step this issue by focusing on the velocity and positions of objects instead. Looking into thematic invariance could be an interesting future direction.

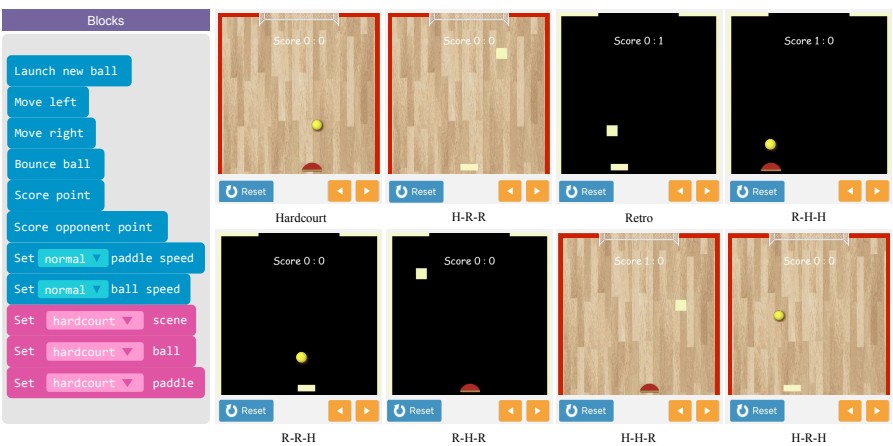

Figure A.2: Bounce can have different "themes" for the background, paddle, and ball. There are two themes to choose from: "hardcourt" and "retro". We show the complete eight different combinations of themes and what the game would look like under these settings.

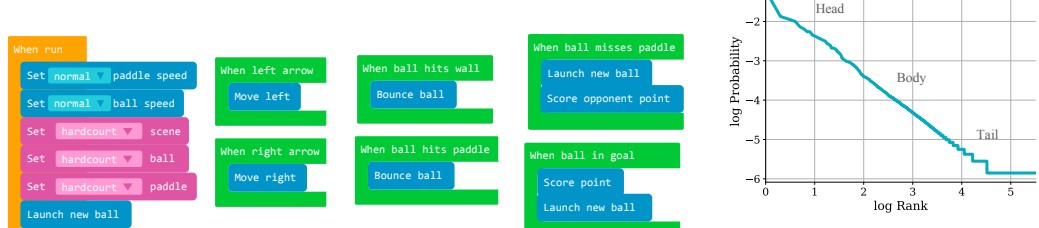

(a) This is the simplest combination that makes up a correct program.

(b) The probability distribution conform to a Zipf distribution.

Figure A.3: This is the drag-and-drop style code editor that students use to create the game Bounce. Conditions such as "when run" or "when left arrow" are fixed in space. Students place commands such as "Score point", under each condition. The submission frequency of each unique program conforms to a Zipfian distribution on a log-log plot.

**Sample student programs**   We show some sample student programs in Figure A.4 to illustrate how complicated the programs can be – even with a limited set of blocks and conditions. Note that even though Figure A.4(a) triggers a visual change when the ball hits the wall, in our current formulation, it does not affect the position and velocity of objects in the game.

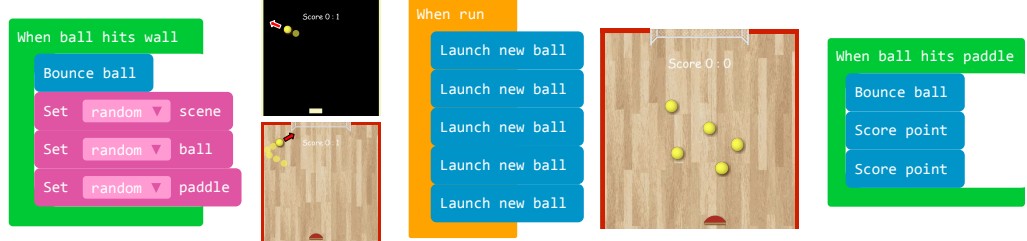

(a) Hit wall change to random theme     (b) Multiple balls when run     (c) Hit paddle wins two points

Figure A.4: We demonstrate three examples of how Bounce can be programmed freely by allowing infinite repetition of commands and commands to be placed under any condition. Only (a) is considered correct since theme change does not affect game play. Both (b) and (c) are considered incorrect. (c) represents a reward design error (give points at incorrect condition). This demonstrates the difficulty of generalization in our setting.

## A.7 Bounce Additional Experiments

### A.7.1 Sampled Evaluation

The experiment result reported in Section 5.2 is from a balanced sampled dataset where we have equal number of correct and broken programs.The underlying submissions are actually imbalanced. We have far more incorrect unique implementations than correct unique implementations. The majority guess (labeling all input programs as broken) would give 86.1% accuracy for body distribution and 94.3% accuracy for the tail distribution. The tail distribution has 66,580 incorrect and 3,999 correct solutions. We show the result in Table 4.

| Contrastive HoareLSTM | Majority Class | Accuracy | Precision | Recall | F1 |
|---|---|---|---|---|---|
| Body-Balanced | 50.0 | 93.4 | 99.5 | 87.2 | 93.0 |
| Body-Sampled | 84.6 | 88.8 | 99.7 | 87.0 | 92.9 |
| Tail-Balanced | 50.0 | 94.0 | 97.4 | 90.4 | 93.8 |
| Tail-Sampled | 92.8 | 94.4 | 100 | 94.0 | 94.4 |

Table 4: We show the precision/recall/F1 for identifying the bug program.

### A.7.2 Ablation on Number of Bug Examples

In our Contrastive HoareLSTM formulation, we assume that teachers will provide a few bug examples. Here we show an ablation study on if we vary the number of provided bug examples, how would it affect our distance-based classifier's performance. We would like to point out that not all broken examples are created equal – some are probably more crucial than others (this could be a great future direction). We simply used our best guess again to choose a smaller set of representatives of broken programs. We did not re-pick the set in any way to optimize their performance. We show the result in Table 5.

| Contrastive HoareLSTM | Accuracy | Precision | Recall | F1 |
|---|---|---|---|---|
| Body-Balanced (3 bug examples) | 50.0 | 50.0 | 100.0 | 66.7 |
| Body-Balanced (5 bug examples) | 89.4 | 99.5 | 79.2 | 88.2 |
| Body-Balanced (7 bug examples) | 92.4 | 99.5 | 85.2 | 91.8 |
| Body-Balanced (10 bug examples) | 93.4 | 99.5 | 87.2 | 93.0 |
| Body-Sampled (3 bug examples) | 84.6 | 84.6 | 100.0 | 91.7 |
| Body-Sampled (5 bug examples) | 84.6 | 84.6 | 100.0 | 91.7 |
| Body-Sampled (7 bug examples) | 86.0 | 99.7 | 83.7 | 91.0 |
| Body-Sampled (10 bug examples) | 88.8 | 99.7 | 87.0 | 92.9 |
| Tail-Balanced (3 bug examples) | 50.0 | 50.0 | 100.0 | 66.7 |
| Tail-Balanced (5 bug examples) | 92.8 | 97.4 | 88.0 | 92.4 |
| Tail-Balanced (7 bug examples) | 93.2 | 97.4 | 88.9 | 92.9 |
| Tail-Balanced (10 bug examples) | 94.0 | 97.4 | 90.4 | 93.8 |
| Tail-Sampled (3 examples) | 92.8 | 92.8 | 100.0 | 96.3 |
| Tail-Sampled (5 examples) | 92.8 | 92.8 | 100.0 | 96.3 |
| Tail-Sampled (7 examples) | 92.8 | 92.8 | 100.0 | 96.3 |
| Tail-Sampled (10 examples) | 94.4 | 100 | 94.0 | 94.4 |

Table 5: We show the precision/recall/F1 for identifying the bug program.