# OpenReview forum: "Play to Grade: Testing Coding Games as Classifying Markov Decision Process"
_NeurIPS.cc/2021/Conference — NeurIPS 2021 Poster_

### Official Review · Reviewer_JNms · 2021-07-09

**Rating:** 7
**Confidence:** 3

**Summary:**

This paper describes a new method for automatically identifying whether a student's code submission for an interactive task (e.g., code for a game with specific rules defined in the assignment) is correct or incorrect. Using a very small training set of at least one correct and one incorrect example for the assignment, each example and student submission is treated as a separate MDP, and goal is to classify whether a student submission is equivalent to the reference MDP for the correct game. To determine if two MDPs are equivalent, they focus on identifying differential states, which are those states that are most different, and then examine whether they exceed a teacher-defined threshold. A "Play to Grade" objective is defined such that maximizing it means finding a policy that frequently reaches differential states. The challenge is then to determine actions to take in the separate MDPs to maximize reaching of differential states, as randomness in the games means that taking the same actions in ech MDP isn't sufficient, as well as determining an appropriate distance function between (state, action) pairs in the new submission and the reference MDP. The paper considers several approaches to both problems: for the distance function, baselines are based on noisy supervised learning and unsupervised learning, and the improved methods examine unexpectedness in the MDP of predicted versus observed states as well as directly contrasting the student submission with an approximated correct MDP and bug MDP. For learning a policy, the paper considers pre-training versus no pre-training, as well as iteratively improving the constant delta that is used within the determination of whether some (s,a) is a differential state. Two types of experiments are performed: a toy example using a car game and a test of real student submissions from Code.org. Results show an improvement over the baselines for the toy example, and an improvement over an approach that treats code-as-text for the Code.org submissions.


**Ethical Concerns:**

I would love to see more attention to whether terms of service like those cited are really equivalent to informed consent, but the research seems to be following standard practice. I am not sure from the documents whether anonymization happens before the researchers ever have access to it (rendering it non-human-subjects) - i.e., stripping of any PII - or if an IRB approved the study as exempt.


**Limitations And Societal Impact:**

The discussion of potential negative impacts seems quite superficial. The tool as created is trying to assess correctness, and the motivation early on is for teachers to know if programs are correct - the distinction that should be made is about the suitability for high-stakes versus low-stakes assessment.

There is also a bridge to be crossed which is not addressed which is moving from correctness feedback to guidance on how to improve. It's not clear from the evaluation whether students are likely to be unaware of their mistakes (in Bounce, it sounds like they would be because games can't be played given the bugs), and if they are not likely to be unaware, then it's not clear that the approach is giving more feedback to them then what they get from trying to play their game. To be clear, I think it's okay that this paper doesn't do everything! But, I would have liked to have seen a little more nuance in discussion of actually applying this, and perhaps references to the kinds of papers noted above that do look at the problem of guiding students.


**Main Review:**

Overall, I think this paper presents a very fun and interesting idea for how to grade interactive programs, and the approach is fairly different from most of the work I'm familiar with. It's less clear to me how generalizable the results are in order to gauge how well this technique is likely to work in practice, and I would have liked to have seen the paper acknowledge a bit more the difference between formative guidance and feedback on correctness versus incorrectness. I also think that some improvements to clarity would increase its impact. In the review below, I address the four points from the reviewer guidelines.

Originality:
While I am not deeply immersed in the background literature here, the approach is novel to the best of my knowledge. With regards to related work, it did seem like a somewhat broader set of work that focuses on automated guidance for programs could be helpful given that that seems to be the ultimate goal with formative feedback, e.g.:
- Rivers, K., & Koedinger, K. R. (2017). Data-driven hint generation in vast solution spaces: a self-improving python programming tutor. International Journal of Artificial Intelligence in Education, 27(1), 37-64.
- Paaßen, B., Hammer, B., Price, T., Barnes, T., Gross, S., & Pinkwart, N. (2018). The Continuous Hint Factory-Providing Hints in Vast and Sparsely Populated Edit Distance Spaces. Journal of Educational Data Mining, 10(1).


Quality:
The paper seems to generally be technically sound. As noted below, there were some points where the lack of clarity made it difficult to evaluate. The major question I had related to the technical work was in the evaluations. My specific concern is how sensitive the algorithm is to the range of incorrect examples and whether the types of errors students tend to make in real environments are likely to be caught by this method. In the toy example, the described error doesn't seem to be generally of the sort we might expect a student to make (it's not clear how clipping the velocity is something that would commonly arise - one might instead expect something to be slightly off in the dynamics through mis-specifying a relationship/formula) and one of the two errors given for bounce again doesn't necessarily seem like something a student is likely to do and to not know is incorrect (having multiple balls when there should only be one). In these examples, it's also not clear how much room there is for creativity or whether the algorithm would be able to identify and not flag differences that are permitted due to "creative discretion", and dealing with creative differences is something that is given as a motivation in the first paragraph of the intro. It seems likely that the way these are accommodated is through the range of sample correct/incorrect submissions containing sufficient information for this to be learned. But, as far as I can tell, the algorithm's sensitivity to the number and range of these submissions is not considered, which may be because in these tasks, there really is only one "correct" MDP or set of behaviors (even if there are many different ways to create code for that behavior). I would have liked to see more discussion of this issue, including potentially evaluating what happens in the real Code.org evaluation is the set of incorrect submissions is varied and minimally, helping the reader understand whether there is any creative discretion permitted (and if so how).

My additional issues are related to both technical quality and clarity. I had a hard time following why the iterative collaborative training was not considered for the Code.org evaluation (as mentioned in lines 326-335). I think the point is that this is not needed, but demonstrating that that would be helpful.

In lines 288-289, it's stated that "We can see that in Table 1, all models including baseline models are able to predict bug states accurately in a single direction agent setting"; Table 1 doesn't seem to imply that for the GMM. In table 2, 95% confidence intervals aren't given like they were for table 1, and additionally, I was wondering why 3 runs are averaged over for table 2 and 5 for table 1. Finally, I would have appreciated some insight on why CT seemed to make GMM worse.

For the code.org experiment, it's not clear how accurate this is likely to be for real student submissions given that they are actually imbalanced, and it also seems misleading to give the table titles as the total number of submissions given that in fact, 500 submissions with 250 from each class are used for each of body and tail. I recognize that this is given in the caption, but even if one wants to report accuracy on the 500 sampled programs, would it also be possible to report accuracy on the fully dataset? For the tail, how many program submissions were correct versus incorrect? I would have also liked to have known a bit more about the range of the actual kinds of errors in the program behavior - a small amount of information is given in 327-328, but better understanding the complexity of the task and the kinds of errors would help with assessing the practicality of this method.

Finally, from what I can tell, the method used for Code-as-text is not given. I assume this is the code-as-text approach cited much earlier in the paper ([11]) as an example of an alternative, but it would be helpful to make this explicit in the evaluation.


Clarity:
There were a number of points where additional clarity would have helped the reader; some of these issues are noted above and some here, and a list of small typos is given at the end of the review. First, I'm not sure that lower case delta is defined precisely, and articulating the relationship between nu and delta would be useful. In the experiments, it's not clear what nu is set to (earlier, it's given as a teacher-set parameter).

There are several other points where clarity about parameter setting would be helpful. For the unsupervised learning baselined, it's stated that "We can adjust sigma based on the performance over D" but I don't believe it's specified how that is done. In 4.2, it's stated that delta can be chosen so that all (s, a) can be labeled as correct. Slightly more explanation here would be helpful, and having more explicitly defined the tuple (s, a) earlier (in 2.1-2.2 - the components are clear, but the way it is being used as a tuple isn't always clear - might help).

In section 4, more clarity about the reward function being used and how this relates to equations 2-3 would be helpful. In paragraph 1, it seems to say that the reward function needs to be about finding bugs (like in 2), but then the second paragraph says the original reward function is good enough. Making more explicit whether this is "in practice, we'll do paragraph 1 using the original reward function, rather than what we just gave" or if something else is meant would be helpful. Additionally, as a limitation, it seems like paragraph 2 is saying that some reward function is needed in the original environment, but then in the discussion, it's stated that this isn't the case. Clarifying this would be helpful.

The compute time and resources are given only in the checklist, not the main text, and the section numbers under 4(d)-(e) are incorrect. The open-sourcing of the dataset is listed as a contribution, but then seems to be more of a plan in 4(b) - a specific timeline would be helpful (i.e., will this definitely be open-sourced if accepted?)


Significance:
If this approach does generalize well to other tasks, I think it has the potential to be very significant, and I think the ideas here are ones that others are likely to build on. The lack of clarity in some places and my uncertainty about whether this is likely to work for a variety of assignments may detract from that significance. Assuming that the dataset is released, as it sounds like is planned, this will be a useful dataset for other researchers.



Typos:
- Line 54: "a students assignment" [missing apostrophe]
- Line 99: "the algorithm takes game objects positional and dynamics as input" [missing word or wrong form (?) and missing apostrophe]
- Lines 132-133: "The difficulty of The challenge "
- Line 148: "direct access to transition function and reward function is not possible" [missing "the"]
- Line 203: "highly reliant on how good our model can approximate the system" ["well" rather than "good"]
- Line 237-238: "The problem with this approach is that the policy is biased toward finishing the original objective in the coding game, it might not lead to any bug discovery" [run-on sentence]
- Line 258: "and reward agent for reaching that state."
- Lines 319-320: "we count any unique program submitted more than 10 submissions as the “body” of the distribution." [missing word?]
- Table 1 caption: "Randomly agent result we report average over 5 runs." [missing word(s) or wrong tense?]


Update after author response:
Thank you for your detailed response! I appreciated especially seeing the breakdown with different numbers of buggy examples, as that seems to help with thinking about how this would apply to use in the classroom. In response to your response on creativity, I definitely understand that this is a difficult problem. I think, however, acknowledging in the text as to how this does and doesn't address creativity, given that it seems to play a central role in your motivation for this work, would be helpful; this is linked to my concerns about clearly acknowledging the limitations, which I think would actually make your work likely to have a larger impact, especially in the communities that are most deeply immersed in the challenges of education and working with students. Thank you also for noting how the velocity clipping is similar to an actual bug - perhaps in the paper a quick mention of how this relates to a real bug would be helpful, in case others (like me) are not imaginative enough to consider how this would arise naturally? Thank you again for your engagement with the reviewing process and for sharing your work.

**Time Spent Reviewing:**

roughly 4 hours, although I did not track closely

---

> ### Author Response · Authors · 2021-08-11
> **Thank you for a thoughtful review. Hopefully we have addressed your concerns and reflected on your suggestions!**
>
> We are grateful for the very detailed review. Thank you for appreciating our idea of grading interactive coding programs as exploring potentially differential states between two MDPs.
>
> Here we address the points you brought up:
>
> > "I would have liked to have seen the paper acknowledge a bit more the difference between formative guidance and feedback on correctness versus incorrectness"
>
> Great suggestion. The need for formative feedback is a key point that has guided our research and is certainly worth bringing up in the paper. Our long term goal is to provide formative feedback -- where we are able to help students learn. This work is a milestone towards that aim. Being able to predict correctness is a quantifiable way to demonstrate progress towards understanding interactive student work.
>
> We will include the suggested literature in our related work section.
>
> > "In the toy example, the described error doesn't seem to be generally of the sort we might expect a student to make (it's not clear how clipping the velocity is something that would commonly arise - one might instead expect something to be slightly off in the dynamics through mis-specifying a relationship/formula)"
>
> The bug in the toy example (clipping velocity) is supposed to emulate a very common mistake that students make in programming Breakout -- it’s called the “sticky paddle” bug where the collision resolution fails and the ball sticks to the paddle and can’t seem to get out. You can search “In Breakout the ball latches itself to paddle and bricks occasionally” in Google.
>
> > "In lines 288-289, it's stated that 'We can see that in Table 1, all models including baseline models are able to predict bug states accurately in a single direction agent setting'; Table 1 doesn't seem to imply that for the GMM."
>
> We misused the word “accurately” in the sentence -- we should have used the word “precisely” instead of “accurately”. The GMM model is able to identify bug states with good precision (60.6 ± 26.0) but low recall (23.8 ± 15.7). Even though we carefully normalized the input for the GMM, the input is not a “learned” representation -- we believe a GMM model can do a good job if we spend more time transforming the input data into a better embedded space. However, since GMM is not our main model, we didn’t spend a lot of energy there, and we believe VAE does a good job highlighting the power of the family of generative models.
>
> > "it's also not clear how much room there is for creativity or whether the algorithm would be able to identify and not flag differences that are permitted due to 'creative discretion'"
>
> Understanding and rewarding creativity is the major challenge in AI for education. Though we didn’t use the Bounce dataset to focus on the problem of understanding creativity, our work opens up an interesting angle to address this very hard challenge. In this work we identified several unique approaches that we believe will be useful in recognizing creativity. The idea of play-to-grade could help to identify the difference between a truly broken solution and one which is playable, but different from the reference solution. Or perhaps we could think of creativity as being related to the delta (distance to the correct solution). We imagine that this line of research will be very exciting future work for both the fields of education and reinforcement learning.
>
> Though we do not explore this in our current setting, we believe a detailed discussion and systematic experiment is warranted for future work.
>
> > "I had a hard time following why the iterative collaborative training was not considered for the Code.org evaluation (as mentioned in lines 326-335). I think the point is that this is not needed, but demonstrating that that would be helpful."
>
> We will include iterative collaboration as a result in the camera-ready version of our paper. For Code.org the results were not substantially improved by using iterative collaboration as compared to play to win. We think that iterative training is an important method as it could be used to recognize edge cases which can’t be identified with play-to-win. In deployment we plan to use a combination of these two models.
>
> > "It's not clear how accurate this is likely to be for real student submissions given that they are actually imbalanced...For the tail, how many program submissions were correct versus incorrect?"
>
> We appreciate the suggestion to sample according to the actual data distribution. We have far more incorrect unique implementations than correct unique implementations. The majority guess (labeling all input programs as broken) would give 86.1% accuracy for body distribution and 94.3% accuracy for the tail distribution. The tail distribution has 66,580 incorrect and 3,999 correct solutions. We thought the balanced representation can provide a better picture of our algorithm. However, it is a great suggestion and we went ahead and re-sampled the data. Here is the result:
>
> (Note that since submission, we found and fixed a small bug in our Bounce simulator code which improved all of our results).
>
> | Contrastive HoareLSTM 	| Majority Class 	| Accuracy 	| Precision 	| Recall 	| F1   	|
> |-----------------------	|----------------	|----------	|-----------	|--------	|------	|
> | Body-Balanced         	| 50.0           	| 93.4     	| 99.5      	| 87.2   	| 93.0 	|
> | Body-Sampled          	| 84.6           	| 88.8     	| 99.7      	| 87.0   	| 92.9 	|
> | Tail-Balanced         	| 50.0           	| 94.0     	| 97.4      	| 90.4   	| 93.8 	|
> | Tail-Sampled          	| 92.8           	| 94.4     	| 100       	| 94.0   	| 94.4 	|
>
> **Table 4:** We show the precision/recall/F1 for identifying the bug program (we have results for non-bug program as well; please let us know if you want to see them)
>
> To answer the question about evaluating on the full dataset. In our current infrastructure, each program takes 1.5s-3s to evaluate. Though not soul-crushingly slow for each assignment, evaluating on the full dataset takes about 2.5-3 days. However, we do not believe this poses a challenge for the deployment, as the new students data would come in as a data stream, and we can scale easily by using standard ML scaling techniques.
>
> > "But, as far as I can tell, the algorithm's sensitivity to the number and range of these submissions is not considered, which may be because in these tasks, there really is only one 'correct' MDP or set of behaviors"
>
> Per your suggestion, we also investigated what if we use less broken example programs. We would like to point out that not all broken examples are created equal -- some probably would be more crucial than others (this would be a great future direction). We simply used our best guess again to choose a smaller set of representatives of broken programs. We didn’t re-pick the set in any way to optimize their performance. Here is the result:
>
> | Contrastive HoareLSTM           	| Accuracy 	| Precision 	| Recall 	| F1   	|
> |---------------------------------	|----------	|-----------	|--------	|------	|
> | Body-Balanced (3 bug examples)  	| 50.0     	| 50.0      	| 100.0  	| 66.7 	|
> | Body-Balanced (5 bug examples)  	| 89.4     	| 99.5      	| 79.2   	| 88.2 	|
> | Body-Balanced (7 bug examples)  	| 92.4     	| 99.5      	| 85.2   	| 91.8 	|
> | Body-Balanced (10 bug examples) 	| 93.4     	| 99.5      	| 87.2   	| 93.0 	|
> | Body-Sampled (3 bug examples)   	| 84.6     	| 84.6      	| 100.0  	| 91.7 	|
> | Body-Sampled (5 bug examples)   	| 84.6     	| 84.6      	| 100.0  	| 91.7 	|
> | Body-Sampled (7 bug examples)   	| 86.0     	| 99.7      	| 83.7   	| 91.0 	|
> | Body-Sampled (10 bug examples)  	| 88.8     	| 99.7      	| 87.0   	| 92.9 	|
> | Tail-Balanced (3 bug examples)  	| 50.0     	| 50.0      	| 100.0  	| 66.7 	|
> | Tail-Balanced (5 bug examples)  	| 92.8     	| 97.4      	| 88.0   	| 92.4 	|
> | Tail-Balanced (7 bug examples)  	| 93.2     	| 97.4      	| 88.9   	| 92.9 	|
> | Tail-Balanced (10 bug examples) 	| 94.0     	| 97.4      	| 90.4   	| 93.8 	|
> | Tail-Sampled (3 examples)       	| 92.8     	| 92.8      	| 100.0  	| 96.3 	|
> | Tail-Sampled (5 examples)       	| 92.8     	| 92.8      	| 100.0  	| 96.3 	|
> | Tail-Sampled (7 examples)       	| 92.8     	| 92.8      	| 100.0  	| 96.3 	|
> | Tail-Sampled (10 examples)      	| 94.4     	| 100       	| 94.0   	| 94.4 	|
>
> **Table 5:** We show the precision/recall/F1 for identifying the bug program (we have results for non-bug program as well; please let us know if you want to see them)
>
> As you can see, with more examples, the performance generally increased. It is difficult to comment on how many incorrect examples are needed -- these would be assignment-dependent. We intentionally chose a very small number of examples in-order to make this solution usable by teachers in the real world, for both small classrooms as well as large online courses.
>
> (continue in the next comment block)

---

> > ### Author Response · Authors · 2021-08-11
> > **(Continued from above)**
> >
> > > "...also liked to have known a bit more about the range of the actual kinds of errors in the program behavior - a small amount of information is given in 327-328, but better understanding the complexity of the task and the kinds of errors would help with assessing the practicality of this method."
> >
> > Based on the additional experiment and comparing the result between balanced evaluation and sampled evaluation, we can conclude that our current algorithm  can misclassify broken solutions as correct solutions -- but not the other way around. We believe this is due to the fact of “novel” bugs, where the novel bug MDP looks more similar to the reference correct MDP but quite different from all the reference bug MDPs.
> >
> > By manually examining the actual programs of these mis-classified examples, they are in general longer, contain behaviors that are not captured in the reference bug MDPs, or have complex transition dynamics that are particularly hard for our (T, R) model to approximate.
> >
> > We will include this discussion in the paper.
> >
> > > "I assume this is the code-as-text approach cited much earlier in the paper ([11]) as an example of an alternative, but it would be helpful to make this explicit in the evaluation."
> >
> > Our “Code-as-text” baseline simply takes code text as input and uses the 10 incorrect programs and 1 correct program as training data to train a supervised learning model. It shows the limitation of supervised learning algorithms that usually require a lot more annotated data.
> >
> > > "For the unsupervised learning baselined, it's stated that "We can adjust sigma based on the performance over D" but I don't believe it's specified how that is done."
> >
> > For an unsupervised learning baseline, the way to know if input (s, a) is what the model has seen before is to compute the reconstruction loss on this input. If the reconstruction loss is a lot higher than the loss observed for training data ($\sigma$), then the input hasn’t been seen. It is provided in the supplementary code file -- we simply set $\sigma$ to the average training loss of the unsupervised learning model (minus the first 100 batches).
> >
> > > In 4.2, it's stated that delta can be chosen so that all (s, a) can be labeled as correct. Slightly more explanation here would be helpful, and having more explicitly defined the tuple (s, a) earlier
> >
> > Adjusting $\delta$ is an important task. One can imagine a delta that is infinitely large, therefore all (s, a) trivially can be classified as correct -- however, such $\delta$ would lead to terrible performance on identifying bug states (because all bug states will be classified as correct). The criteria we use to select $\delta$ is: assign $\delta$ to be the smallest possible value that is able to classify 99% of (s, a) pairs as correct for the correct reference MDP.
> >
> > > "I would love to see more attention to whether terms of service like those cited are really equivalent to informed consent, but the research seems to be following standard practice. I am not sure from the documents whether anonymization happens before the researchers ever have access to it (rendering it non-human-subjects) - i.e., stripping of any PII - or if an IRB approved the study as exempt."
> >
> > Thank you for bringing up such an important issue. We take ethical conduct very seriously in our research. The user agreement of Code.org (which you can access online) specified that they are able to share anonymized data for research purposes for the benefit of the students’ future learning experiences. Their engineers have cleaned the dataset before they were shared with us. The data we have access to only contain a randomly generated user ID as well as the actual program. There is no comment or personal identifiable information in our data. We are glad to have this opportunity to address it and will include this in the appendix of our paper.
> >
> > Thank you for providing suggestions and engaging in this discussion with us. We are happy to explain more if needed.

---

### Official Review · Reviewer_CvhS · 2021-07-15

**Rating:** 6
**Confidence:** 4

**Summary:**

The paper formalizes the a binary classification problem: whether a student-authored, interactive code program is correct or not. The difficulty of this problem stems from the interactive nature of the program: in order to determine whether it is correct, the algorithm must "play" with it (because it is interactive) and then determine whether it is correct by finding whether the "play" results in some undesired/buggy behaviors. The proposed solution consists of an agent that learns to play with the program, and a modified LSTM (the so called hoareLSTM) that classifies programs (more specifically, certain "state" and "action" pairs in a program) into correct or incorrect. Evaluation on 2 simulated environments and on 1 program assignment demonstrate the effectiveness of the proposed approach over a few baselines (3 in the simulated environment and 2 in the program assignment).

**Limitations And Societal Impact:**

See review above.

**Main Review:**

The paper explores an interesting and important problem. The proposed solution is interesting. However, at this paper's current state, I do not recommend acceptance. My main concern is the lack of clarify and details. I also have some concerns on the real-world applicability of the proposed method. Some of the issues I listed below may seems insignificant on its own, but combined, they make it a bit difficult to get the full picture and understanding of the proposed method. It would have been nice if more details were available in the supplementary material.


1.  The notion of a "game" is not super clear to me. Is it coding environment itself, in which leaners are completing the programming assignment on code.org? Or is it the output of the programing assignment, i.e., what the students are asked to do in the programming assignment? Or is it the environment in which the proposed agent is trying to identify the "bug-triggering state"? Some graphical illustrations or explanations about the "game", the coding environment of code.org, and example solutions would be very helpful.


2. The definitions of action and state in the program MDP is not super clear to me. For example, what are the representations of states and actions, i.e., are they vectors representing something? What is the action space, and are actions continuous or discrete? Are the set of states finite? What do actions and states correspond to in the context of programs? Again, some illustrations would be helpful.


3. The argument that prior work on code-level does not generalize well beyond 10 lines of code (line 74 - 76) is not quite well substantiated. First, it's not clear how many time steps the program MDPs are; if they are more or less 10 steps, then the proposed method is no better than previous methods that work on 10 lines of code. Some additional characterization and statistics on the code.org dataset would also be helpful, e.g., the average lengths $t$ of the programs etc. From the supplementary material, the programs seem rather short. Second, I don't see how the proposed method "sidesteps the complexity of static code analyses" because it seems the interactive/dynamic code analysis introduces even more challenges.


4. The educational utility of identifying the "bug-triggering state" is not demonstrated. Without this, it is difficult to recognize the educational value of this work, because the whole purpose of this work is to give feedback. It would be great if an illustrative example can be provided, i.e., at a certain state the program produces undesirable behavior, and thus is considered wrong. I acknowledge the code.org dataset is still proprietary, but is it possible for the authors to create a fake program on code.org by themselves (I assume code.org is generally accessible) and demonstrate your trained algorithm using your custom-made example?


5. It would be great if additional details and analyses of the proposed method are available. For example:
- How are the teachers' gold solution(s) and "best guesses" of wrong solutions (lines 84 - 87) used? Are the gold solutions the "reference MDPs"? If so then how are the best guessed wrong solutions used?
- In implementing algorithm 1: how many trajectories are collected for the different environment? Is the number related to the complexity of the environment? How is $\delta$ adjusted and why it started at 0.1?
- Why the hoare triple predictor necessarily needs to be an auto-regressive model? It seems like a feedforward network might also work because the loss in (6) is on a single pair of $(s,a)$? And by the way, only in (4) do $a$ and $s$ have the $t$ subscripts and it's a bit confusing whether $(s,a)\sim\pi$ is sampling a single $(s,a)$ or a sequence of them.
- Why Table 1's setting (5 runs with CI) is different from Table 2's setting (3 runs without CI)? More details on the experimental details would be helpful.


6. It seems that the evaluation on the code.org data is on a single assignment. If this is true, how does this method generalize to other assignment, if one needs to collect a large number of students' submissions and train the method for each programming assignment? Some discussions on the scalability, generalizability, and applicability of the proposed method in the real-world would be helpful.


7. There are a number of typos:
- line 35: "for a 20 seconds"
- line 54: "with a students assignment"
- line 106: "fewer from"
- line 154: "solving is task is"
- line 184: "they are also been extended"
- line 196: "if we learn to"
- line 224: "sample MDPs effective"
- line 238: run-on sentence
- line 136: I don't think you can write the probability of $D(M,M') > \nu$ because I thought $D(M,M')$ is not a random variable but just a number, so it does not have a distribution?



=========
post rebuttal:
Having seen the discussions and reviews from the other reviewers and the authors rebuttal, the value of this paper outweigh the many presentation and clarity issues. I have changed my rating and hope the final version is much more polished than the submitted one.

**Time Spent Reviewing:**

9

---

> ### Author Response · Authors · 2021-08-11
> **Thank you for a thoughtful review. Hopefully we have addressed your concerns and reflected on your suggestions!**
>
> Thank you for the thoughtful feedback and appreciation that this is an interesting and important problem. We appreciate the suggestions, and the concerns that you raised. We believe a large portion of the writing problems can be fixed by adding more precise details in our paper.
>
> We also hope to also assure you of the real world applicability of this work as we think that this is one of the strengths of our research. Our primary goal is to make AI that can be directly used to help the students who are learning to program online for free -- and relative to many successful NeurIPS papers, we believe we have achieved results which are especially close to being relevant to the real world. While we present a theoretically interesting problem, it is grounded in the needs of the students in code.org. Our next step is to make a tool, based on this work, that we can provide for free to students. Importantly, along with this work we will be publishing the code.org dataset of student submissions to Bounce. By doing so we provide other researchers both the ability to (1) verify the applicability of our research, and (2) to build upon our ideas. We believe that the play-to-grade challenge is a great opportunity for the AI community to support work that could directly benefit learners.
>
> You brought up many helpful suggestions which we will use to improve the clarity of our work. We address the points that you brought up, in order:
>
> 1. The game refers to the output of the code (i.e., when a student is asked to write a program, and such a program can be executed as a playable game). There are many examples like Bounce on Code.org. We explained this in Line 94 to 96 -- instead of using the word “implementation of the interactive assignment”, we can use “output of the interactive assignment” instead.
> 2. The state of the program MDP is the velocity, position of objects in the game. Action is the allowable action an agent can take to control the object in the game -- i.e., which direction to apply acceleration for the car (described in line 268-269); or control paddle to go left or right (which should seem intuitive in Figure 2). We can make these more explicit in text.
> 3. We believe you might be confused about the definition of our MDP. Our MDP is not trying to “recreate” the program code line by line. The MDP we construct is trying to train an agent to play the actual game (the output of the program) and expose bugs along the way. The student can write 10 lines of program or 100 lines of program -- the underlying game will be the same (if the game is implemented correctly). The programming assignment Bounce we analyze does have shorter length code, but even with such short length, the “code-as-text” baseline still struggled because of the few-shot nature of the challenge. We want the algorithm to work on novel assignments. When only trained on 11 examples (1 positive and 10 negative code files) it performs poorly. Because the student programs form a long-tail distribution, static code analysis solutions need a lot more annotated data; while a behavior-driven solution needs a lot less. We believe by actually “playing” each student’s solution, we are creating an algorithm that is much closer to the human style of verification (i.e., human testers often play a alpha/beta version of the game to find bugs -- they do not read the code).
> 4. We believe “Identifying a bug state” holds tremendous educational utility. For beginner programmers, it is important for them to see where the program went wrong and try to develop a solution by themselves. By identifying a bug state, we can record a 2-3 seconds short video around the bug state and display to students where their program made a mistake. We plan to collaborate with Code.org to deploy our trained algorithm in a real production environment, but this live experiment is beyond the scope of our technical paper.
> 5. Thank you for bringing up these clarifying questions!
>     - For Line 84-87, our Contrastive HoareLSTM uses both gold solution and best guesses of wrong solutions as described in Line 206-208. Implementation details or experiment details such as number of trajectory can be found in the notebook.
>     - In Bounce, we keep a buffer of 6400 trajectories to train our classifiers. We sample about 19200 trajectories in total -- which is not a large dataset to train sequence models. Delta does not need to start at 0.1 -- it can start anywhere and be adjusted by following the procedure Line 250-252. We will write separately about these details in the appendix if the paper is accepted.
>     - Hoare triple predictor does not need to be an auto-regressive model -- it can be a Markov feedforward model as well, as you suggested. We chose LSTM because it’s a common choice among the sequence modeling community and it does not assume Markov independence between states when modeling the environment. As for (a, s) ~ π, this is a shorthand notation commonly used in RL for a much more complicated sampling -- more realistically, all (a, s) have subscript t.
>     - We apologize for the inconsistent run times -- 5 runs were chosen for Table 1 because we saw much greater variability in model performance between each run, so we wanted to shrink the confidence interval and provide more reliable statistics for the readers. 3 runs were chosen for Table 2 because the variability is a lot smaller. We will increase Table 2 run to 5 runs and report CI for it as well.
> 6. Play-to-grade is few shot by design so that it can be used in classrooms small and large. You are correct that the real world evaluation was from the assignment Bounce on code.org data which is an exceptionally large data set. We chose to use such a large data set as it allowed for a thorough evaluation of our algorithm. Code.org is unique in that the hundreds of thousands of solutions provide an extensive evaluation of our algorithm in the long tail of student submissions. However, play-to-grade only needs a single reference solution and a handful of reference errors in order to train. We plan to use future versions of this algorithm in courses with only a few hundred students as well as in large online platforms.
> 7. We appreciate the catching of these typos. We will correct these in our paper.
>
> We hope these answers provide some explanations to address your concerns and showcase that our work is solving a significant challenge in a satisfying manner. Perhaps you will be convinced by the other reviewers who found the work to be a solid contribution to the body of research presented at NeurIPS.
>
> Thank you for providing suggestions and engaging in this discussion with us. We are happy to explain more if needed.

---

> > ### Comment · Reviewer_CvhS · 2021-08-27
> > **Thanks for the detailed additional info**
> >
> > Those cleared most of my confusions about the work. I now tend to concur with the other reviewers. Hopefully the authors can make appropriate changes so the paper is clear and accessible for researchers in the NeurIPS community and beyond.

---

> ### Comment · Reviewer_TaJm · 2021-08-18
> **Value of identifying buggy states**
>
> CvhS questions to the educational utility of identifying the "bug-triggering state." Without the ability to find a bug-triggering state (to this reviewer's knowledge) there is no other evaluative feedback given to students on their submissions to code.org. Going from zero feedback to some feedback is already useful, and then to have that feedback take the form of a detailed animation of how their game can go wrong goes much further.

---

### Official Review · Reviewer_TaJm · 2021-07-17

**Rating:** 8
**Confidence:** 4

**Summary:**

The paper addresses a societally relevant and technically challenging problem by addressing it in very general terms (rather than offering a solution specific to one application domain). Both the problem and the solution are exceptionally well motivated.

The key conceptual contribution is to identify student submitted programs as MDPs (formalizations of environments for interaction in which certain outcomes are associated with rewards/utility). Although the motivated goal is to provide students with (rich?) feedback on their program, this is reduced to binary grading: classification (although any feedback is better than none in this setting where feedback is desperately needed).

The resource contribution of a dataset is also appreciated.

**Ethical Concerns:**

None.

**Limitations And Societal Impact:**

The assumption that all student programs have a common action space feels appropriate, but the assumption that they all have the same state space feels much more limiting. It might not occur in the Code.org setting, but it feels quite realistic for some student programs to track state elements (such as the health of enemy characters) where others do not, even when given the same prompt to design their games/programs. The assumption of shared state is critical for being able to transfer policies between the many related submissions being graded. But an approach that did not need to train policies to access diverse trajectories might bypass this limitation.

The authors are encouraged to use their specific knowledge of the domain to explain why even though having different amounts of variable state in different programs might be very common in general, it is almost entirely absent in their domain.

**Main Review:**

The identified problem is highly significant, but the proposed solution is somewhat less. It is unclear if the empirical result achieved is mostly a result of better state space exploration (which can be achieved without involving reinforcement learning) or if the quite complicated collaborative training step (involving all of DQN) was essential. That said, identifying this problem setting as a few-shot classification task to be solved by similarity search is quite interesting.*

* It also suggests a form of rich feedback for students. When students see their grade, they could also be shown which of the reference solutions theirs was most similar to (and even see one of the reference trajectories for that environment, showing how the same kind of bug manifests in their program).

The central challenge searching for diverse states in interactive software to uncover bugs feels inherently much more related to the areas of automated exploration (e.g. the upstream literature of Go-Explore https://arxiv.org/abs/1901.10995 or recent applications of RRT-based methods to games http://ceur-ws.org/Vol-2313/KEG_2019_paper_9.pdf), and input fuzzing (e.g. upstream of https://www.ei.ruhr-uni-bochum.de/media/emma/veroeffentlichungen/2020/02/27/IJON-Oakland20.pdf) than reinforcement learning. Exploration/fuzzing is often *unmotivated* (not based on maximizing rewards) or at *intrinsically* motivated (not dependent on environmentally-provided rewards). It might be the case that something like Go-Explore (which might be seen as a kind of randomized breadth first search using screenshot pixels to judge distinct states) or a classic fuzzer (given access to an abstraction of the game’s state vector) can provide the diverse trajectory samples needed for the proposed classifiers to work just as well without defining any rewards or running DQN.


**Time Spent Reviewing:**

2

---

> ### Author Response · Authors · 2021-08-11
> **Thank you for a thoughtful review. We addressed your concerns and reflect on your suggestions!**
>
> Thank you for this thoughtful review.
>
> We appreciate the recognition of the potential real-world impact our task entails!
>
> > "[This work] also suggests a form of rich feedback for students. When students see their grade, they could also be shown which of the reference solutions theirs was most similar to (and even see one of the reference trajectories for that environment, showing how the same kind of bug manifests in their program)."
>
> This is a great suggestion and we plan to do exactly this in order to give useful feedback to students. We look forward to measuring the learning gains that tools, based on the play-to-grade algorithm, enable. Visualizing results in a way that is most helpful to students will be a core part of this continued project.
>
> > "The central challenge searching for diverse states in interactive software to uncover bugs feels inherently related to the areas of automated exploration"
>
> The papers you included are helpful in situating our work within literature. We will make sure to include them.
>
> > "The assumption that all student programs have a common action space feels appropriate, but the assumption that they all have the same state space feels much more limiting"
>
> This is a very good point. In future work we plan to explore alternative methods that rely only on pixel space to ensure same state representation for all student programs. In the context of Code.org, which is typical in many intro to programming assignments, the state spaces have a natural structure which we were able to leverage. We are excited to extend our results to more complex environments. In the meantime, there are heuristic engineering approaches to conform more open-ended assignments to a consistent state space -- such as capturing state modifications at the API level.
>
> We are happy to discuss ideas, or answer more questions if they come up!

---

### Comment · Area_Chair_x8rJ · 2021-09-13
**About dataset as contribution - seeking Authors' input**

Dear Authors,

Thank you for your detailed responses, which helped in answering most of the reviewers' questions. As we are concluding our discussions, we want to relay an additional concern that has come up in our discussions. This is related to questions on open-sourcing the dataset that the reviewer(s) asked earlier, but it seems to have not been answered in the authors' responses so far.

The main concern here is that the paper lists one of the four contributions as "We will release a dataset of over 700k submissions to support further research." In the checklist, it is mentioned that the data is currently proprietary, making it sound more of a plan without a clear timeline. Given that the reviewers consider this dataset contribution in the reviewing process, a specific timeline is important.  The questions where we want to get the authors' input are: "Do the authors already have permission from code.org to open-source it? What is the specific timeline that the authors have in their plan to open-source it? Does this timeline aligns with the conference (e.g., camera-ready deadline?)"

The main point of concern is that code.org has not released open-sourced datasets in the past few years, even though their proprietary dataset has been used in recent paper(s).

We will be thankful if the authors can provide some input on the above questions.

---

> ### Author Response · Authors · 2021-09-14
> **Data Release**
>
> Hi,
>
> We have already obtained written permission from Code.org to release this dataset several months ago during our communications and initial receiving of the dataset. We will make sure the data see the light of day. Unlike previous proprietary datasets, this dataset only contains student programs without including any comments and personal information. We did extensive cleaning and standardization to make sure the data are safe to publish.
>
> We plan to release it at the camera-ready deadline, as suggested.
>
> Authors

---

> > ### Comment · Area_Chair_x8rJ · 2021-09-14
> > **Re: Data Release**
> >
> > Thanks for the response.

---

> > > ### Author Response · Authors · 2021-09-18
> > > **Email from Code.org**
> > >
> > > We received some communication from Code.org
> > > The release of dataset isn't going to be an issue.
> > >
> > > ---------- Forwarded message ---------
> > > From: Baker Franke <baker@code.org>
> > > Date: Fri, Sep 17, 2021, 8:27 PM
> > > Subject: All clear
> > > To: Authors
> > >
> > > Hey there,
> > >
> > > Hadi + Lawyer + COO + head of product see no need for any further process for you sharing the data set as part of publication.  You’re all clear.

---

### Decision · Program_Chairs · 2021-09-28

**Decision:**

Accept (Poster)

**Comment:**

The paper studies the problem of automatically identifying whether the student program for an interactive coding task is correct or not. The proposed method is based on a novel idea of mapping a student program to an MDP and then developing an RL agent that can play in this MDP with the objective of classifying it as correct or incorrect. Experiments are performed on a synthetic task as well as on a programming task from code.org. The reviewers acknowledged the importance of the studied problem and the potential impact in the domain of coding education. The reviewers raised several concerns in their initial reviews. The reviewers appreciated the authors' responses, which helped in answering most of their questions.

One specific concern that came up is regarding releasing the dataset. The paper lists one of the four contributions as "We will release a dataset of over 700k submissions to support further research." In the checklist, it is mentioned that the data is currently proprietary, making it sound more of a plan without a clear timeline. Given that the reviewers consider this dataset contribution in the reviewing process, a timeline is important. It is expected that the authors already have a plan in place and would open-source the dataset during the conference timeline (e.g., camera-ready deadline) if the paper is accepted.

Overall, the reviewers have a positive assessment of the paper. The reviewers have provided detailed feedback in their reviews, and we strongly encourage the authors to incorporate this feedback when preparing a revised version of the paper.


**Consistency Experiment:**

NeurIPS has a long history of experimentation. In 2014, NeurIPS ran an experiment in which 10% of submissions were reviewed by two independent committees to quantify the randomness in the review process. This year, we repeated a variant of this experiment to see how the quality of the review process has changed over time.  This paper was part of the experiment and was therefore assigned to two committees (consisting of reviewers, an Area Chair, and a Senior Area Chair) that reached independent decisions.  If both committees made the same recommendation, this recommendation was followed. If a single committee recommended acceptance, the paper was accepted (with the exception of a few cases in which the other committee identified what we considered a fatal flaw, e.g., an error in a key result).

Both committees reached the same decision: **Accept (Poster)**

The other committee assigned to the paper recommended **Accept (Poster)**.  You can find the other set of reviews, along with any follow up discussion with the authors here:
https://openreview.net/forum?id=hjBEEXWNFH3